# The Variability of Vitamin D Concentrations in Short Children with Short Stature from Central Poland—The Effects of Insolation, Supplementation, and COVID-19 Pandemic Isolation

**DOI:** 10.3390/nu15163629

**Published:** 2023-08-18

**Authors:** Joanna Smyczyńska, Natalia Pawelak, Maciej Hilczer, Anna Łupińska, Andrzej Lewiński, Renata Stawerska

**Affiliations:** 1Department of Pediatrics, Diabetology, Endocrinology and Nephrology, Medical University of Lodz, 90-419 Lodz, Poland; 2Department of Endocrinology and Metabolic Diseases, Polish Mother’s Memorial Hospital—Research Institute in Lodz, 93-338 Lodz, Poland; pawelak.natalia@gmail.com (N.P.); maciej.hilczer@umed.lodz.pl (M.H.); anna.lupinska@umed.lodz.pl (A.Ł.); andrzej.lewinski@umed.lodz.pl (A.L.); renata.stawerska@umed.lodz.pl (R.S.); 3Department of Pediatric Endocrinology, Medical University of Lodz, 93-338 Lodz, Poland; 4Department of Endocrinology and Metabolic Diseases, Medical University of Lodz, 93-338 Lodz, Poland

**Keywords:** vitamin D, sun exposure, SARS-CoV-2 pandemic, lockdown, supplementation

## Abstract

The aim of the study was to investigate the effects of seasonal variability of insolation, the implementation of new recommendations for vitamin D supplementation (2018), and the SARS-CoV-2 pandemic lockdown (2020) on 25(OH)D concentrations in children from central Poland. The retrospective analysis of variability of 25(OH)D concentrations during the last 8 years was performed in a group of 1440 children with short stature, aged 3.0–18.0 years. Significant differences in 25(OH)D concentrations were found between the periods from mid-2014 to mid-2018, from mid-2018 to mid-2020, and from mid-2020 to mid-2022 (medians: 22.9, 26.0, and 29.9 ng/mL, respectively). Time series models created on the grounds of data from 6 years of the pre-pandemic period and used for prediction for the pandemic period explained over 80% of the seasonal variability of 25(OH)D concentrations, with overprediction for the first year of the pandemic and underprediction for the second year. A significant increase in 25(OH)D concentrations was observed both after the introduction of new vitamin D supplementation guidelines and during the SARS-CoV-2 pandemic; however, the scale of vitamin D deficiency and insufficiency was still too high. Time series models are useful in analyzing the impact of health policy interventions and pandemic restrictions on the seasonal variability of vitamin D concentrations.

## 1. Introduction

The main role of vitamin D is influencing the calcium-phosphate metabolism, and its long-term deficiency is important in the pathogenesis of rickets in children and bone mineralization disorders in adults. Apart from that, it has many other proven or highly probable roles, including the regulation of the proliferation and differentiation of normal and cancerous cells [1,2,3], reducing the risk of cardiovascular diseases [4] and of some cancers, as well as the modulation of the immune system, including protective effects against both autoimmune diseases and infections. Proper vitamin D levels reduce the risk of viral infections, including the SARS-CoV-2 virus [5].

The primary source of vitamin D in humans is its synthesis in the skin, requiring sufficient exposure to UVB radiation, the source of which is sunlight. However, it is insufficient not only in our latitude but also in countries with higher insolation, at least during the autumn and winter [6,7,8,9]. Another important source of vitamin D is food products. Despite the well-documented seasonality of serum 25(OH)D concentrations, it is difficult to directly assess the amount of vitamin D provided by the sun or food; nevertheless, the scale of vitamin D deficiency in central European countries is high [10]. As it has previously been documented, vitamin D intake from food alone is insufficient to achieve adequate concentrations in the human body, even in terms of abundant sunshine [11,12]. Hence, an additional supply of vitamin D seems necessary. The solutions of this problem include direct vitamin D supplementation and the fortification of food products with vitamin D [13,14].

Vitamin D is a group of fat-soluble prohormones which can be synthesized naturally in the human body from 7-dehydrocholesterol to cholecalciferol (vitamin D3) that depends on sunlight exposure or may be provided through the dietary supplementation of ergocalciferol (vitamin D2) and vitamin D3. To achieve metabolic effects, vitamins D2 and D3 must be converted into active forms through the hydroxylation reactions. The result of the first hydroxylation in liver is calcifediol—25-hydroxyvitamin D [25(OH)D] that is subsequently hydroxylated in kidneys to calcitriol—1,25-dihydroxyvitamin D [1,25(OH)2D]. Calcitriol synthesis is limited by the accessibility of calcifediol, which is a substrate. A serum concentration of 25(OH)D is considered the main marker of vitamin D supply. The biological effects of vitamin D are mediated by the vitamin D receptor (VDR). Current knowledge concerning the sources of vitamin D in humans and the pleiotropic effects of this vitamin has been summarized by Bouillon et al. [15].

To date, literature reviews indicate vitamin D deficiency in the general population worldwide [16,17]. Its concentration depends on many components, such as age, sex, race, latitude, and season [18]. Vitamin D has pleiotropic effects and, for this reason, correcting its deficiency through proper supplementation is important for public health. Grant et al. [19], in the 2022 narrative review, discussed which vitamin D concentrations were appropriate for various health outcomes, such as cardiovascular diseases, hypertension, cancers, type 2 diabetes mellitus, and many others, which were the most common causes of death. Optimal thresholds for the different effects of vitamin D ranged from 25 ng/mL to 60 ng/mL. In this paper, a special section has been devoted to the importance of vitamin D for the course of SARS-CoV-2 infections. The significance of vitamin D during the COVID-19 pandemic and the pleiotropic effects of vitamin D have also been discussed during the fifth International Conference “Vitamin D—Minimum, Maximum, Optimum”, held in Warsaw, Poland, in October 2021 [20].

In Poland, the guidelines for vitamin D supplementation for Central Europe, published in 2013 [21], were used for several years. This did not solve the problem of the high prevalence of vitamin D deficiency in the Polish population. Chlebna-Sokół et al. [22], in 2016, found vitamin D deficiency in more than 70% of children referred to the hospital for symptoms suggesting bone metabolism disorders. The study involved the entire developmental period from neonates to 18 years of age. They also observed an increase in the prevalence of vitamin D deficiency with age. A high incidence of insufficient vitamin D concentrations together with their seasonal variability in children not suffering from the symptoms typical for skeletal disorders, assessed in 2014–2018, was also reported in the previous paper of our research group, published in 2019 [23]. 

In 2018, Rusińska et al. [24] published the guidelines for vitamin D supplementation for the Polish population that were widely promoted among medical staff. According to these recommendations, vitamin D should be supplemented in healthy children and adolescents when its availability from natural sources is limited (due to insufficient sun exposure and insufficient dietary supply). In addition, there were specific recommendations for using a higher vitamin D dosage in the groups at risk of vitamin D deficiency. The expected effect of implementing these recommendations should be an improvement in vitamin D supply and reduced prevalence of vitamin D deficiency, especially in autumn and winter.

During the initial phase of the COVID-19 pandemic lockdown, the possibility of sun exposure was restricted, which must have caused a reduction in vitamin D synthesis in the skin during the spring and summer of 2020 at the population level. On the other hand, during the SARS-CoV-2 pandemic, vitamin D supplementation has been recommended in response to numerous reports of its effect on the prevention of viral infections [5,25,26,27].

The aim of the study was to assess the effects of:the seasonal variability of insolation,implementing new guidelines of vitamin D supplementation,the pandemic situation with limited sun exposure due to lockdown and strong recommendations to increase vitamin D supplementation on serum 25(OH)D concentrations in children.

## 2. Materials and Methods

The retrospective analysis included 1440 children (879 boys, 561 girls), aged 3.0–18.0 years, with short stature, i.e., height SDS below −2.0, according to Polish reference standards [28], admitted to a single tertiary reference center in Poland, and diagnosed from January 2014 to the end of June 2022. In all children, 25(OH)D serum concentrations were measured on the second day of hospitalization, fasting, in morning hours. None of the patients had recommended therapeutic doses of vitamin D; however, they could supplement this vitamin as over the counter (OTC) medications and consume foods fortified with vitamin D. The patients with disorders of thyroid function, pituitary hormone disorders (except for ones with isolated growth hormone deficiency), disorders of adrenal function, hyperthyroidism or uncompensated hypothyroidism, and any calcium-phosphorus imbalance and/or impaired PTH secretion, as well as those diagnosed with diseases that may influence vitamin D supply (including coeliac disease, other malabsorption syndromes, anorexia nervosa, and chronic kidney disease, etc.) were excluded from the study. 

The concentrations of 25(OH)D in the serum were measured with electrochemiluminescence binding assay (ECLIA), Roche, standardized against LC-MS/MS, with a range of detection of 5.0–100.0 ng/mL. According to the current guidelines for the Polish population [24], vitamin D deficiency was defined as serum 25(OH)D concentrations below 20 ng/mL, suboptimal concentrations—20–30 ng/mL, optimal—30–50 ng/mL, high—50–100 ng/mL, and toxic—over 100 ng/mL.

First, the patients were divided into four groups with low, suboptimal, optimal, and high 25(OH)D concentrations. As only one child had a 25(OH)D concentration over 100.0 ng/mL, this patient was included in the subgroup with high vitamin D levels. 

Next, all the patients were classified with respect to the year and season of hospitalization. In the guidelines of Rusińska et al. [24], published in May 2018, prophylactic vitamin D supplementation is, in general, recommended for all children, apart from those with sufficient sun exposure in the period from May to September, thus the effects of the application of these rules should be noticeable from the third quarter of 2018. The SARS-CoV-2 pandemic started in Poland in March 2020; however, its possible effects on vitamin D supply seemed to be delayed, as median serum 25(OH)D concentrations in the second quarter of 2020 and 2021 were very similar (22.5 ng/mL vs. 22.6 ng/mL). So, 25(OH)D concentrations were compared between the patients diagnosed from mid-2014 to mid-2018, from mid-2018 to mid-2020, and from mid-2020 to mid-2022, and the groups were labelled as “Old Guidelines”, “New Guidelines”, and “Pandemic”, respectively. The patients diagnosed in the first and second quarter of 2014 were excluded from this part of the analysis. The selection for the analysis of full 4-year and 2-year periods allowed us to avoid a bias related to the unequal representation of data collected in different seasons. For the purpose of this study, it was assumed that the seasons corresponded to the appropriate quarters of the year.

For statistical analysis, a Shapiro–Wilk test was used first for the assessment of the distribution of serum 25(OH)D concentrations in the studied group and in particular subgroups. Due to the lack of a normal distribution of 25(OH)D levels, a non-parametric Kruskal–Wallis test was applied for comparisons between particular groups, followed by a post hoc Bonferroni–Dunn test, if applicable. Statistical significance was defined as *p* < 0.05. 

The second part of the analysis involved creating models of the seasonal (quarterly) variability of serum 25(OH)D concentrations based on time series regression, including the data on insolation during the observation period and testing the effect of the SARS-CoV-2 pandemic. The onset of pandemic isolation was in March 2020, and many restrictions of outdoor activities and online education lasted during 2021, while the vaccination of children at least 12 years of age started in June 2021, and for those aged 5–11 years in December 2021. So, it might be assumed that pandemic isolation could decrease the sun exposure of Polish children from spring 2020 to the end of 2021. The first model included data concerning median 25(OH)D concentrations from the last 6 years of the pre-pandemic period (from spring 2014 to winter 2020) and was used for predictions of 25(OH)D concentrations in the pandemic period (starting from spring 2020). The second model included additional data concerning insolation during the 3 months preceding the measurements of serum 25(OH)D concentrations in particular children, as it had previously been documented that vitamin D concentrations correlated with insolation in previous months [23]. The information concerning the number of sunny hours in particular months during the study period and 3 months before, starting from October 2013 to June 2022, was obtained from the website of the Institute of Meteorology and Water Management in Poland (https://klimat.imgw.pl/pl/climate-maps/#Sunshine/Monthly accessed on 27 April 2023). All the models were created in quarterly intervals that seemed to reflect the seasonal variability of 25(OH)D concentrations (the creation of monthly models was abandoned due to their low readability and a relatively low number of patients diagnosed in some months). The assessment of the fit of model forecasts to real data was used to estimate the impact of the analyzed interventions (the introduction of New Guidelines and the SARS-CoV-2 pandemic) on serum 25(OH)D concentrations.

## 3. Results

As distributions of both serum 25(OH)D concentrations and of patients’ age were different from normal distribution, both in the whole studied group and in particular subgroups, median values and interquartile (25–75 centile) ranges are presented. In the studied group, the median 25(OH)D concentration was 24.0 ng/mL, with an interquartile range of 18.3–30.2 ng/mL. Only 25.1% of measured 25(OH)D values were within the normal range, 41.9% were suboptimal, 32.0% confirmed vitamin D deficiency, and only 1.0% were high. In general, vitamin D concentrations were higher in younger children, with significant differences in the patients’ age between the subgroup with low serum 25(OH)D concentrations and all the remaining subgroups, as well as between the subgroups with optimal and suboptimal 25(OH)D levels. The basic characteristics of the studied group and of particular subgroups with respect to 25(OH)D concentration are presented in Table 1. 

In the studied group, 25(OH)D concentrations were the highest in children assessed from mid-2020 to mid-2022, and lowest in those assessed from mid-2014 to mid-2018. See Table 2; for raw data see Figure 1.

The seasonal variability of serum 25(OH)D concentrations in particular years and seasons is shown in Figure 2. 

Non-parametric tests for multiple comparisons followed by post hoc comparisons showed that all differences in serum 25(OH)D concentrations between the groups diagnosed in particular time periods (i.e., “Old Guidelines”, “New Guidelines”, and “Pandemic”) were significant (*p* < 0.05). Further comparisons of serum 25(OH)D concentrations between these groups, for seasons in particular, also demonstrated significant differences for winter, spring, and autumn, but not for summer; see Figure 3. 

Detailed data on previous insolation for particular years and months, calculated according to the daily numbers of sunny hours in the 3 months preceding the assessment of serum 25(OH)D concentration, are shown in Table 3. Significant correlations between 25(OH)D concentrations and insolation in previous months were observed, with the best one being between the median serum concentration of 25(OH)D and the cumulative number of sunny hours during the previous 3 months (r = 0.695, *p* < 0.05).

Concentrations of serum 25(OH)D in particular seasons before and after the implementation of the New Guidelines of vitamin D supplementation and during the SARS-CoV-2 pandemic.

Finally, the models of the seasonal variability of serum 25(OH)D concentrations were created on the grounds of data from 6 years of the pre-pandemic period (from spring 2014 to winter 2020) and used for prediction for the pandemic period (from spring 2020 to mid-2022) in order to assess the influence of the SARS-CoV-2 pandemic on vitamin D supply (i.e., to validate fitting the model to the pandemic situation). The first model of median 25(OH)D concentrations in particular seasons (quarters) explained 84.0% of its variability. The second model, including the additional variable “Previous Insolation” (see Table 3), explained 88.6% of median 25(OH)D concentration variability. Interestingly, both models overpredicted 25(OH)D concentrations in spring, summer, and autumn 2020, and underpredicted in 2021 and the first two quarters of 2022; however, the differences between real and forecasted values were insignificant. The model including the variable “Previous Insolation” is presented in Figure 4. Implementing the detailed data on insolation added less than 5% accuracy to the model based only on the seasonal variability of serum 25(OH)D concentrations. It should also be noted that, during the whole study period, except for summer 2021, the median values of 25(OH)D concentrations were below the normal range (i.e., below 30 ng/mL). 

## 4. Discussion

In our study, a seasonal variability in vitamin D concentrations was confirmed, with almost 75% incidence of suboptimal and low serum 25(OH)D concentrations during the whole study period. Similar conclusions have been drawn from other papers published in recent years. Mean serum 25(OH)D concentrations were higher during the summer–autumn seasons compared to the winter–spring seasons [12,29,30,31,32]. The highest prevalence of serum 25(OH)D below 20.0 ng/mL in Greek adults was found in the spring season, precisely in March, by Dimakopoulos et al. [33]. Basińska-Lewandowska et al. [34] compared only two seasons—spring and autumn—in Polish adults and obtained mean serum 25(OH)D concentrations at the levels of 18.1 ± 7.37 ng/mL and 24.58 ± 7.72 ng/mL, respectively. Seasonal differences in the prevalence of vitamin D deficiency or sufficiency observed in that study were highly significant. 

During the first days of the COVID-19 pandemic, the authorities in many countries imposed restrictions that included limiting going out of the house (a so-called lockdown). One of the consequences of this situation was reduced sun exposure and, as a result, a greater prevalence of vitamin D deficiency. Rustecka et al. [35] evaluated the effect of staying home during the pandemic on vitamin D concentrations among Polish children. They compared serum 25(OH)D concentrations between two groups of patients who had blood samples taken either before the pandemic (January 2019 to February 2020) or during the first pandemic year (March 2020 to February 2021). Among children over 1 year of age, the mean vitamin D concentration was significantly lower during the pandemic than in the pre-pandemic period (35 ± 18 ng/mL and 31 ± 14 ng/mL, respectively), while in infants, serum 25(OH)D levels were normal. Moreover, season-dependent changes in vitamin D levels were observed in the pre-lockdown period, while no such changes were observed during the lockdown. In our study, the obtained serum 25(OH)D concentrations are lower than those reported by Rustecka et al. [35], and we did not observed such a decrease in 25(OH)D concentrations during the first year of the pandemic; however, a direct comparison of the obtained results is difficult as we reported median 25(OH)D levels. Nevertheless, serum 25(OH)D concentrations in 2020 were lower than those predicted in the model based on the data from the pre-pandemic period, but they surpassed the forecast since 2021.

Tsugawa et al. [36] provided a study conducted on young women in which they measured 25(OH)D concentrations from May 2016 to June 2017 and in September 2020 (after lockdown due to COVID-19). They showed a significant difference in 25(OH)D levels between the samples obtained in September 2016 and in September 2020 (21.7 ± 6.6 ng/mL vs. 13.2 ± 5.0 ng/mL). Additionally, serum 25(OH)D concentrations showed seasonality with the highest values in September. Similarly, in our study, the highest 25(OH)D concentrations were observed each year in the summer season.

Jastrzębska et al. [37] demonstrated changes in vitamin D concentrations among 24 young soccer players over the course of a year. The study started in September 2019 and ended during the COVID-19 pandemic in August 2020. Significant differences in serum 25(OH)D concentrations between the seasons were reported, with the lowest concentrations in autumn and winter, and during the home isolation period in spring 2020.

Lippi et al. [38] compared vitamin D concentrations in the outpatient population before and after the first day of lockdown (10 March 2020). The results showed higher vitamin D concentrations and a modest but lower likelihood of vitamin D deficiency in the first 9 months of the pandemic (from 10 March 10 to 11 December 2020) than in the same period of the previous 2 years. The authors linked the paradoxical rise in serum 25(OH)D concentrations at the start of the COVID-19 pandemic and during the consequent lockdown to the higher proportion of males who were tested while compared to the same period in the previous two years. In general, higher vitamin D concentrations in males than in females have been confirmed in other studies [11,27,30], which may indeed offer an explanation for these findings. 

Ferrari et al. [39] investigated vitamin D concentrations among patients in different age groups in 2019 and 2020. In their study, serum 25(OH)D concentrations were higher during the lockdown period than in the same period a year earlier. The authors did not find a direct link between vitamin D concentrations and sun exposure, but indicated that different variables, such as vitamin D supplementation, may have influenced this. Conversely, Beyazgül et al. [40] reported a decrease in vitamin D in Turkish school-aged children and adolescents in first year of the pandemic with respect to the pre-pandemic period. Li et al. [41] observed significantly higher vitamin D levels in 2020 than in 2019. In Chinese children, however, in February, March, and April 2020 vitamin D concentrations were lower than in the same months of 2019. In our study, as in most other reports, significantly higher 25(OH)D concentrations during the SARS-CoV-2 pandemic than in the pre-pandemic period were observed. Moreover, seasonal variability related to differences in sun exposure was an important variable in the created models of 25(OH)D levels. 

In the context of the SARS-CoV-2 pandemic, very recent studies concerning the relationships between solar activity or solar cycles and epidemics seem to be especially interesting [42,43,44]. The direct effects of weather variables on SARS-CoV-2 transmission have also been confirmed in studies conducted in different countries [45,46,47,48]. In the Spanish population, a direct influence of higher insolation on a lower rate of COVID-19 spread has been documented [49]. Even in Brazil, which is a tropical country in which the pandemic onset was during the summer, high solar radiation presented to be the most important climatic factor suppressing the spread of COVID-19 [50]. There are also suggestions that vitamin D may be the link between these phenomena [43]. This hypothesis may be supported to some extent by a very recent observation of Polish authors that the risk of COVID-19 infection was increased in subjects with severe 25(OH)D deficiency (below 12 ng/mL) [51]. 

These issues were not directly analyzed in our study; however, in the studied population of Polish children, the pandemic lockdown resulted in decreased sun exposure which resulted in lower vitamin D concentrations with respect to the prediction based on insolation. 

In our study, seasonal differences in insolation turned out to be the variable that explained 84% of the seasonal variability of serum 25(OH)D concentrations. In fact, we did not assessed the individual dietary habits of our subjects and their possible seasonality; however, there is some evidence from other studies that vitamin D intake from diet in children is below the recommended level [52,53], even in terms of vitamin D fortification school meal programs [53]. A study comparing Brazilian women living in different latitudes showed an almost-twice-as-high basal serum 25(OH)D concentration in women living in latitudes with high ultraviolet B (UVB) exposure (16° S) than in those living in latitudes “without UVB exposure” (51° N), despite no difference in vitamin D intake from diet between them and with similar 25(OH)D increase in terms of supplementation of 15 μg cholecalciferol in both groups [54]. For our considerations, the studies on the population of Canada, Alberta (the region located at a similar latitude as Poland) seem to be especially important, documenting low vitamin D intake and high prevalence of its insufficiency both in pregnant women and in children, with no significant difference in serum 25(OH)D concentrations in 3-month-old infants measured in summer and in winter [55,56]. We have not found studies indicating the seasonality of vitamin D intake in food, especially for higher vitamin D content in the diet in summer with respect to winter, not only in Polish children but also in other populations. Moreover, as in Poland vitamin D supplementation in healthy children and adolescents has been recommended for October to April [24], it seems that its seasonal application could only weaken the observed effects related to insolation.

The consumption of dietary supplements has increased in recent years. Especially during the COVID-19 pandemic, this related to vitamin D supplementation. This was most likely due to emerging reports on the effect of vitamin D supplementation on reducing the risk of SARS-CoV-2 infection, disease severity, and risk of death [57]. Puścion-Jakubik et al. [58] conducted a survey on the consumption of dietary supplements, particularly vitamin D and zinc, among Polish adults during the three waves of the COVID-19 pandemic. The results indicated that the largest percentage of respondents used supplements containing vitamin D. In addition, it was shown that the consumption of dietary supplements was significantly higher among those with higher medical education, indicating a high awareness of the health-promoting aspects of supplementation among this group. 

Our study was not devoted directly to the assessment of vitamin D supplementation; however, it documented a significant increase in serum 25(OH)D concentrations both after the implementation of the New Guidelines of vitamin D supplementation in 2018 [24] and during the SARS-CoV-2 pandemic. Nevertheless, the scale of vitamin D deficiency and insufficiency is still too high and further efforts seem necessary to improve the vitamin D supply. On the other hand, there is a need to prevent the cases of uncontrolled overdosing of vitamin D. A very recent update of the guidelines for preventing and treating vitamin D deficiency in Poland [59] includes recommended vitamin D dosing in different age groups and in specific clinical situations (such as pregnancy and lactation, prematurity, overweight and obesity, chronic diseases, and diets), together with the determination of maximum doses of cholecalciferol in the general population by age for the prophylaxis of vitamin D deficiency.

## 5. Conclusions

A significant increase in serum 25(OH)D concentrations has been observed after the introduction of new vitamin D supplementation guidelines, which indicates the effectiveness of the actions to implement these recommendations. A further increase in 25(OH)D concentrations during the SARS-CoV-2 pandemic seems to be related to increased vitamin D supply with the intention to reduce the risk of COVID-19 infection. Nevertheless, the scale of vitamin D deficiency and insufficiency among children is still too high. Time series models have proven to be useful in analyzing the impact of health policy interventions and pandemic restrictions on the seasonal variability of vitamin D concentrations.

## Figures and Tables

**Figure 1 nutrients-15-03629-f001:**
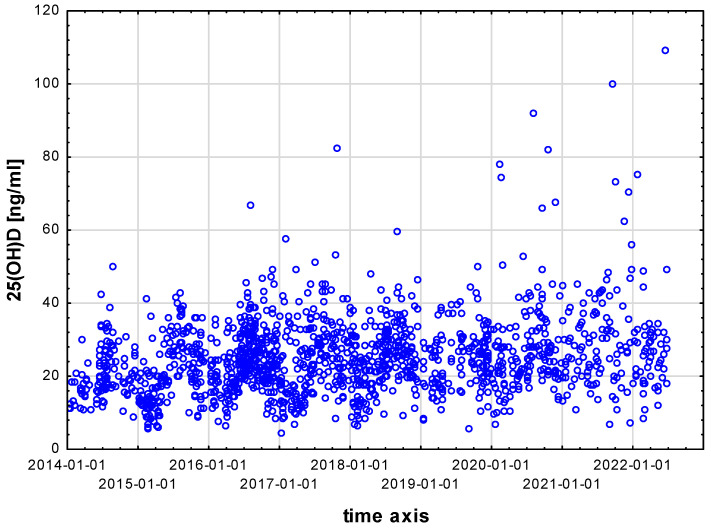
Serum 25(OH)D concentrations in particular patients during the study period.

**Figure 2 nutrients-15-03629-f002:**
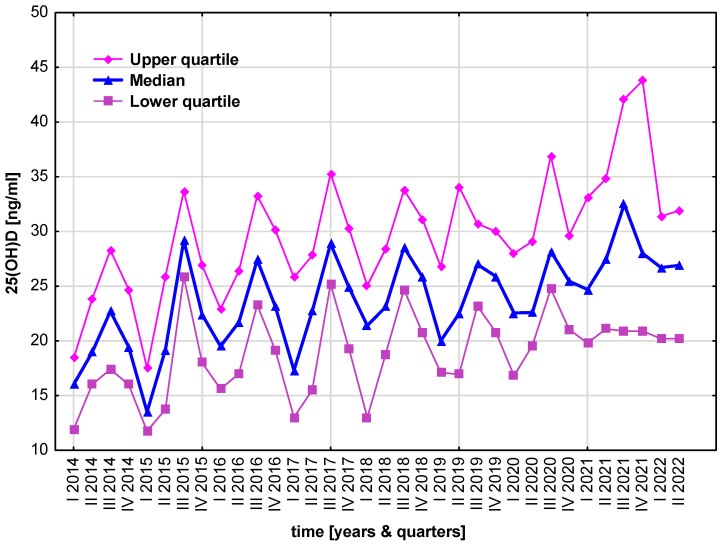
Variability of serum 25(OH)D concentrations in particular years and seasons during the study period.

**Figure 3 nutrients-15-03629-f003:**
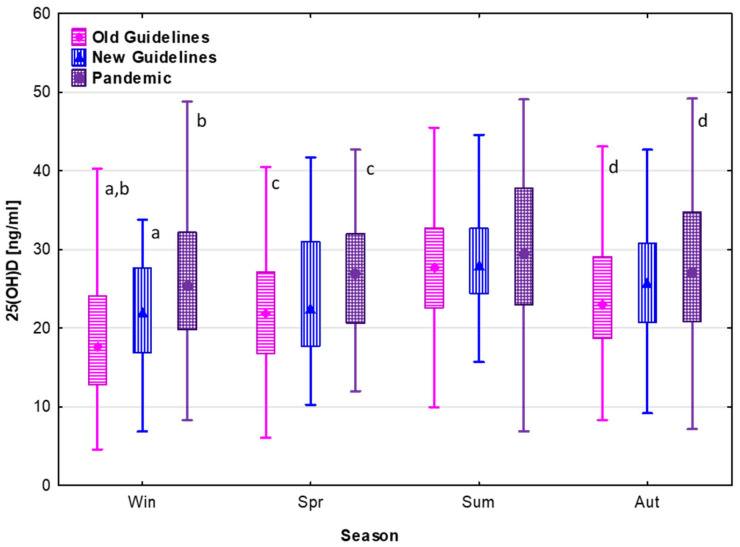
Serum 25(OH)D concentrations are presented as median (point), 25–75 centile (box), and non-outlier range (whiskers); significant differences: a, b, c, d—*p* < 0.05.

**Figure 4 nutrients-15-03629-f004:**
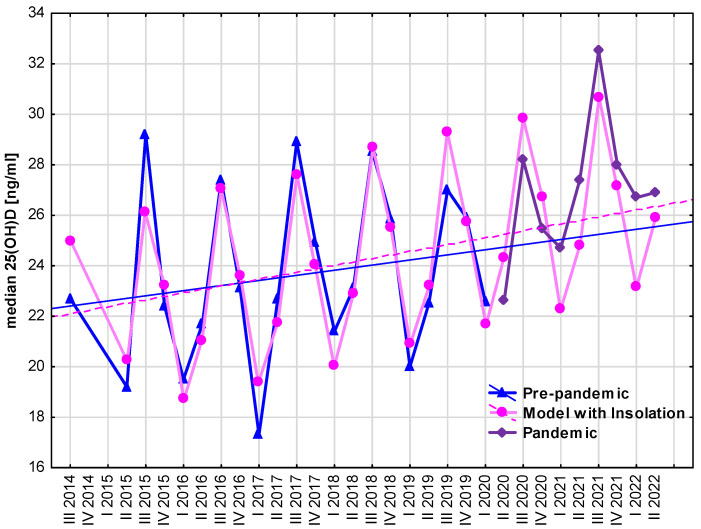
Model of median 25(OH)D concentrations in the pre-pandemic period based on quarterly seasonality and previous insolation, with the forecast for the pandemic seasons.

**Table 1 nutrients-15-03629-t001:** The number and age of patients in the whole studied group and in particular subgroups with respect to serum 25(OH)D concentrations.

Group	All	Low	Suboptimal	Optimal	High
No of patients (boys/girls)	1440 (879/561)	461 (285/176)	604 (362/242)	361 (223/138)	14 (9/5)
Age [years] median (lower-upper quartile)	10.0	11.1 ^a,b,c^	10.3 ^a,d^	8.6 ^b,d^	10.6 ^c^
	(6.9–12.9)	(8.5–13.6)	(7.1–12.8)	(5.5–12.0)	(6.2–11.3)

Significant differences: ^a^, ^b^, ^d^—*p* < 0.001, ^c^—*p* < 0.05.

**Table 2 nutrients-15-03629-t002:** Comparison of serum 25(OH)D concentrations before and after the implementation of the New Guidelines of vitamin D supplementation.

Group	All	Mid-2014 to Mid-2018 (Old Guidelines)	Mid-2018 to Mid-2020 (New Guidelines)	Mid-2020 to Mid-2022 (Pandemic)
No of patients (boys/girls)	1397 (854/543)	841 (501/340)	314 (195/119)	242 (158/84)
Age [years] median (lower-upper quartile)	10.0 (7.0–12.9)	10.1 (7.1–13.1)	9.8 (6.8–12.6)	9.9 (7.0–12.7)
25(OH)D [ng/mL] (lower-upper quartile)	24.2	22.9 ^a,b^	26.0 ^a,c^	29.9 ^b,c^
	(18.6–30.3)	(17.3–28.7)	(20.0–30.7)	(21.3–34.3)

Significant differences: ^a^, ^b^, ^c^—*p* < 0.05.

**Table 3 nutrients-15-03629-t003:** Previous insolation, calculated for the patients diagnosed in particular years and seasons as mean number of sunny hours in the 3 months preceding the assessment of serum 25(OH)D concentrations.

	2014	2015	2016	2017	2018	2019	2020	2021	2022
Winter	2.4	1.6	2.2	2.0	1.7	2.1	2.2	1.8	2.1
Spring	5.0	4.3	4.3	4.3	5.5	4.2	5.3	4.5	5.6
Summer	6.8	8.0	8.5	7.9	9.0	8.6	8.0	8.2	-
Autumn	5.6	7.3	6.2	5.2	7.4	5.9	6.6	5.7	-

According to the data from https://klimat.imgw.pl/pl/climate-maps/#Sunshine/Monthly, accessed on 27 April 2023.

## Data Availability

The data presented in this study are available on request from the corresponding author.

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
