# Peer review of "The Variability of Vitamin D Concentrations in Short Children with Short Stature from Central Poland—The Effects of Insolation, Supplementation, and COVID-19 Pandemic Isolation"

_nutrients, 2023, doi:10.3390/nu15163629_

Round 1
Reviewer 1 Report
Thank you for submitting your paper to the Nutrients journal. The topic carries potential in this field. However, this reviewer has identified several issues with the paper as detailed below.
1. The title may need to be rephrased, as the study design is insufficient to fully investigate the effects of insolation and supplementation, particularly. For instance, the authors sourced the insolation data from a website, not from individual subject data. As for supplementation, it is unclear whether the subjects adhered to the guidelines published in 2018.
2. The authors have not evaluated the dietary status of the subjects. Given that the serum vitamin D concentration can be influenced by a subject's diet, this oversight presents a major issue in the paper. As it stands, we cannot accurately determine the impact of seasons, insolation, or guidelines on serum 25(OH)D concentration.
3. L42-43: The authors state "it is insufficient in our latitude, at least during the autumn and winter, hence, an additional supply of this vitamin is necessary." This assertion requires relevant citations.
4. L44-46: The authors claim "food alone is insufficient to achieve adequate concentrations in the human body, so proper supplementation is essential [6]." This statement may need to be tempered. If this is the case, your data should show that most subjects fall into the Low group for 25(OH)D concentration.
5. L100: The authors' reasons for selecting subjects with short stature are unclear.
6. This reviewer recommends the use of "serum 25(OH)D concentration" instead of merely "25(OH)D," especially in the figures and tables. This is to ensure that the figures and tables are comprehensible independently from the main text.
Author Response
Dear Reviewer,
Thank you for your detailed revision and comments. The manuscript has been modified according to these suggestions.
- The title of the paper
We are fully aware of the fact that we’ve not assessed either insolation or vitamin D supplementation in particular patients (especially with respect to their adherence to guidelines). Instead, we wanted to assess, if there were any general trends that could be related to insolation (seasonal, disturbed by pandemic lockdown) or to increased vitamin D supplementation (year-to-year). The title of the paper reflects these assumptions. Of course, we do not deny the potential impact of individual differences sun exposure and in vitamin D supplementation on the results obtained in particular children, however the study evaluated a large group of over 1,400 patients, so these individual differences should be of relatively minor importance.
- Evaluation of dietary status of the subjects
We can fully agree that serum vitamin D concentration can be influenced by a subject's diet. However, there we observed clear seasonal variability of 25(OH)D concentrations and “Previous insolation” explained over 85% of this variability. This finding, of course, does not exclude the possible effect of other seasonal factors on 25(OH)D levels. There is no data that vitamin D supply in diet might be higher during summer than in winter, moreover, vitamin D supplementation is generally recommended outside the summer period. So, we are convinced that seasonal fluctuations of median 25(OH)D concentrations are related above all to insolation. Unfortunately, our study is a retrospective one and it is impossible to complete more details. As we’ve stated in section “Materials and Methods” (lines 103-104), we know that none of the patients had recommended therapeutic doses of vitamin D.
- The problem of insufficient vitamin D synthesis in skin (L42-43)
Indeed, this issue has not been studied by us. Reference [6] (Kiely & Black, 2012) refers to the whole paragraph, not only to the last sentence. However, we have found and added other references that support the statement concerning insufficient vitamin D synthesis in the skin (Cashman et al., 2016; O’Neil et al., 2016).
- The problem of food insufficiency to maintain optimal vitamin D levels (L44-46)
As reported in Table1, in our study 1065 out of 1440 children had low or suboptimal 25(OH)D serum concentrations that supports the statement that proper vitamin D supply is not achieved by the majority of subjects. We consider adequate vitamin D concentrations only at “optimal” but not at “suboptimal” level, so, under that assumption, our data confirm the need for vitamin D supplementation. Moreover, we have found some studies that clearly show the difficulties or even impossibility to achieve optimal vitamin D supply by food alone. So, we decided to expand the References by few papers (Vierucci et al., 2014; Oliosa et al., 2023; Manios et al., 2018; Hayes & Cashman, 2017; Cashman, 2015) and added more detailed comments.
- Reasons for selection children with short stature (L100)
We decided to retrospectively assess patients with short stature as the relatively high group of children who underwent detailed diagnosis (the same procedures in each case during 8 years of data collection) and fulfilled the same inclusion and exclusion criteria (L103-111). In general, we have recruited children in whom the only health problem was short stature. In fact, we had no data on comparable group of healthy children with normal height for such long-term analysis. However, our findings concerning seasonal variability of 25(OH) concentrations and its changes related to SARS-CoV-2 pandemic are in line with the results of other studies. To our best knowledge, this is the first study in which time series models were used for the detailed analysis of both seasonal and long-term variability of vitamin D concentrations.
- Thank you for the comment concerning using term “serum 25(OH)D concentration" not only “25(OH)D” – we have verified the text accordingly.
We hope that we managed to meet your expectations and to improve the quality of the paper.
Kind regards,
Joanna Smyczyńska
Reviewer 2 Report
Smyczyńska et al investigate the effects of seasonal variability of insolation, implementation of new recommendations for vitamin D supplementation and SARS-CoV-2 pandemic lockdown on 25(OH)D concentrations in children from central Poland. It seems original and weel structured.
As indicated in the "Instructions for authors" included on the website I recommend introducing section 5. CONCLUSIONS.
Author Response
Dear Reviewer,
thank you for a revision of our paper and recommendations. Section 5 "Conclusions" is added according to your suggestions.
Kind regards,
Joanna Smyczyńska
Reviewer 3 Report
In this manuscript, Smyczyńska et al. present the study on variability of vitamin D concentrations in children – effects of insolation, supplementation, and COVID-19 pandemic isolation.
The authors examined the effects of seasonal variability of insolation, implementation of new recommendations for vitamin D supplementation (2018) and SARS-CoV-2 pandemic lockdown (2020) on 25(OH)D concentrations in children from central Poland.
Using time series models on the ground of data from 6 years of pre-pandemic period and prediction for pandemic period explained over 80% of seasonal variability of 25(OH)D concentrations, they found that significant increase of 25(OH)D concentrations was observed both after the introduction of new vitamin D supplementation guidelines and during the SARS-CoV-2 pandemic, however the scale of vitamin D deficiency and insufficiency is still too high. The results indicated that Time series models are useful in analysing the impact of health policy interventions and pandemic restrictions on seasonal variability of vitamin D concentrations. This study is interesting, and the experiments are well thought out and executed. The authors should improve the writing skills before its publication.
In this manuscript, Smyczyńska et al. present the study on variability of vitamin D concentrations in children – effects of insolation, supplementation, and COVID-19 pandemic isolation.
The authors examined the effects of seasonal variability of insolation, implementation of new recommendations for vitamin D supplementation (2018) and SARS-CoV-2 pandemic lockdown (2020) on 25(OH)D concentrations in children from central Poland.
Using time series models on the ground of data from 6 years of pre-pandemic period and prediction for pandemic period explained over 80% of seasonal variability of 25(OH)D concentrations, they found that significant increase of 25(OH)D concentrations was observed both after the introduction of new vitamin D supplementation guidelines and during the SARS-CoV-2 pandemic, however the scale of vitamin D deficiency and insufficiency is still too high. The results indicated that Time series models are useful in analysing the impact of health policy interventions and pandemic restrictions on seasonal variability of vitamin D concentrations. This study is interesting, and the experiments are well thought out and executed. The authors should improve the writing skills before its publication.
Author Response
Dear Reviewer,
thank you for all comments. According to the suggestions, what might be improved, we have verified and modified the references and reworked the description of some results. Moreover, the text has been checked once more to correct language errors.
Kind regards,
Joanna Smyczyńska
Round 2
Reviewer 1 Report
1. The title of the paper
The authors stated, "we wanted to assess if there were any general trends related to insolation (seasonal, disturbed by pandemic lockdown) or increased vitamin D supplementation (year-to-year)." While this is understood, the appropriateness of the title and abstract in conveying this to Nutrient readers is questionable. It seems you evaluated insolation and guideline compliance in the population. Moreover, referencing "a large group of over 1,400 patients" might be misleading. In this study design, a sample size of over 1,400 subjects isn't necessarily large. Therefore, claiming that "these individual differences should be of relatively minor importance" based on this size alone is not justified. If you wish to make this assertion, it should be supported by scientific evidence.
2. Evaluation of dietary status of the subjects
You mentioned, "we observed clear seasonal variability in 25(OH)D concentrations and 'Previous insolation' explained over 85% of this variability." This is solely based on the variables you utilized in your analysis. Had you incorporated dietary data, the outcome might have been entirely different. I strongly recommend that you provide references to justify why dietary status might exert a minimal influence on the results presented in your paper.
3. Reasons for selection children with short stature
This reviewer understands the reasons for selection children with short staturethe. This reviewer recommends that the authors should include "short stature" in the title.
Author Response
Dear Reviever,
Thank you once more for your detailed comments and suggestions. The paper has been modified in accordance to these recommendations.
We’ve extended the title of the paper, adding the information that our study relates to short children from central Poland. We have not put a suggestion concerning the sample size into the text of the manuscript, as we really have no data of dietary habits of our patients and are not able to assess its effect.
We’ve added to references few papers showing insufficiency of vitamin D intake in food for maintaining optimal serum 25(OH)D concentrations (Borger et al., 2021; Calvo et al., 2022; Mendes et al., 2021; Aghajafari et al., 2018; Letorneau et al., 2022), however we could not find the studies devoted to seasonality of vitamin D intake in food.
Added text is highlighted in green.
Kind regards,
Joanna Smyczyńska